# Unraveling the rate-determining step of C$_{2+}$ products during electrochemical CO reduction

Wanyu Deng [1,2,3], Peng Zhang [1,2], Yu Qiao[3], Georg Kastlunger [3], Nitish Govindarajan[3], Aoni Xu [3], Ib Chorkendorff [3], Brian Seger [3] ✉ & Jinlong Gong [1,2] ✉

The electrochemical reduction of CO has drawn a large amount of attention due to its potential to produce sustainable fuels and chemicals by using renewable energy. However, the reaction's mechanism is not yet well understood. A major debate is whether the rate-determining step for the generation of multi-carbon products is C-C coupling or CO hydrogenation. This paper conducts an experimental analysis of the rate-determining step, exploring pH dependency, kinetic isotope effects, and the impact of CO partial pressure on multi-carbon product activity. Results reveal constant multi-carbon product activity with pH or electrolyte deuteration changes, and CO partial pressure data aligns with the theoretical formula derived from *CO-*CO coupling as the rate-determining step. These findings establish the dimerization of two *CO as the rate-determining step for multi-carbon product formation. Extending the study to commercial copper nanoparticles and oxide-derived copper catalysts shows their rate-determining step also involves *CO-*CO coupling. This investigation provides vital kinetic data and a theoretical foundation for enhancing multi-carbon product production.

Electrochemical reduction of CO (COER) into high-value fuels and chemicals is envisioned as a promising path toward storing renewable electricity in chemical bonds[1–5]. Copper (Cu)-based catalysts are the widely accepted materials that can effectively electrochemically catalyze the formation of valuable multi-carbon (C$_{2+}$) products with reasonable selectivity and activity[6–8]. However, its electrochemical performance still has deficiencies that are providing barriers for its commercialization[9–11]. An in-depth understanding of the reaction mechanism can guide the improvement of their activity and selectivity, as well as improve the energy efficiency and economics of the entire electrolyzer.

Recently, many mechanistic studies have been reported relating to reaction intermediate species[12–16], reaction paths[17–20], reaction rate-

determining steps (RDSs)[13,21–23], and the influence of electrolytes[24–26]. The identification of the RDS, as the most critical mechanism guide, still lacks comprehensive and convincing experimental evidence. Three mechanisms have been generally proposed as the potential RDS through density functional theory calculation, including *CO-CO(g) coupling[13,27], *CO-*CO coupling[21,22,28–30], and the protonation of *CO to *CO(H)[20,28,29,31,32]. Recent kinetic data has shown that the first mechanism is highly unlikely to be the RDS because the current density of C$_{2+}$ is a constant in the CO partial pressure range of 0.6 to 1.0 bar[21], and this mechanism would suggest it should be first order in high CO pressure. According to Li et al., the *CO-*CO coupling was favored as RDS due to a Tafel slope of 118 mV·dec$^{-1}$, CO reaction order ($n_{CO}$) of 0 at higher CO pressure, and pH independence of C$_{2+}$ current density at electrolytes

[1]Key Laboratory for Green Chemical Technology of Ministry of Education, School of Chemical Engineering and Technology, Tianjin University, Collaborative Innovation Center of Chemical Science and Engineering (Tianjin), Tianjin 300072, China. [2]Joint School of National University of Singapore and Tianjin University, International Campus of Tianjin University, Binhai New City, Fuzhou 350207, China. [3]Department of Physics, Technical University of Denmark, 2800 Kgs Lyngby, Denmark. ✉e-mail: brse@fysik.dtu.dk; jlgong@tju.edu.cn

**Table 1 | Summary of proposed reaction schemes with (*)CO coupling as the possible RDS for C$_{2+}$ products formation (detailed derivations are shown in Supplementary Note 1)**

| Step[a] | Proposed reaction scheme for C$_{2+}$ products formation | CO pressure | $n_{CO}$ | $n_{H^+}$ | $n_{H_2O}$ |
|---|---|---|---|---|---|
| A1 | $CO + * \rightarrow {}^*CO$ | | | | |
| A2 | ${}^*CO + {}^*CO + e^{\delta-} \xrightarrow{RDS} {}^*C_2O_2{}^{\delta-} + *$ | High | 0 | 0 | 0 |
| | | Low | 2 | 0 | 0 |
| B1 | $CO + * \rightarrow {}^*CO$ | | | | |
| B2 | ${}^*CO + CO + e^{\delta-} \xrightarrow{RDS} {}^*C_2O_2{}^{\delta-}$ | High | 1 | 0 | 0 |
| | | Low | 2 | 0 | 0 |

[a]A$_i$ and B$_i$ represents the reaction step of $i$ in mechanism A and B, respectively.

**Table 2 | Summary of proposed reaction schemes with *CO protonation as the possible RDS for C$_{2+}$ products formation (detailed derivations are shown in Supplementary Note 1)**

| Step[a] | Proposed reaction scheme for C$_{2+}$ products formation | CO pressure | $n_{CO}$ | $n_{H^+}$ | $n_{H_2O}$ |
|---|---|---|---|---|---|
| C1 | $CO + * \rightarrow {}^*CO$ | | | | |
| C2 | ${}^*CO + H_2O + e^- \xrightarrow{RDS} {}^*CO(H) + OH^-$ | High | 0 | 0 | 1 |
| | | Low | 1 | 0 | 1 |
| D1 | $CO + * \rightarrow {}^*CO$ | | | | |
| D2 | ${}^*CO + H^+ + e^- \xrightarrow{RDS} {}^*CO(H)$ | High | 0 | 1 | 0 |
| | | Low | 1 | 1 | 0 |
| E1 | $CO + * \rightarrow {}^*CO$ | | | | |
| E2 | $H_2O + * + e^- \rightarrow {}^*H + OH^-$ | | | | |
| E3 | ${}^*CO + {}^*H \xrightarrow{RDS} {}^*CO(H) + *$ | High | −1 | 1 | 0 |
| | | Low | 1 | 1~−1[b] | 0 |
| F1 | $CO + * \rightarrow {}^*CO$ | | | | |
| F2 | $H^+ + * + e^- \rightarrow {}^*H$ | | | | |
| F3 | ${}^*CO + {}^*H \xrightarrow{RDS} {}^*CO(H) + *$ | High | −1 | 1 | 0 |
| | | Low | 1 | 1~−1 | 0 |

[a]C$_i$, D$_i$, E$_i$, and F$_i$ represent the reaction step of $i$ in mechanism C, D, E and F, respectively.
[b]1 ~ −1 represents that the value is ≥ −1 and ≤1, and all the other expressions are similar.

with pH ranging from 7 to 13[30]. This mechanism was further supported by the recent theoretical-experimental study by Kastlunger et al., where they found pH-independent activity in both alkaline and acidic conditions[33]. However, a more detailed analysis revealed the $n_{CO}$ was -1 at lower CO pressures[31,34]. This was consistent with the RDS of *CO protonation to *CO(H), but it cannot explain the pH independence of the C$_{2+}$ products current density in acidic conditions. Thus, a consistent and convincing RDS still needs further experimental exploration and analysis. Although previous studies have undertaken similar investigations, no convincing results have been obtained thus far. The primary reason for this is the incomplete analysis of dynamics, leading to insufficient experimental design and erroneous data analysis, among other factors, which have resulted in extensive controversies in this field[31,34].

In this study, the RDS of COER to C$_{2+}$ products was investigated by combining and comprehensively analyzing theoretical derivations and kinetic experimental results. Based on various previously proposed RDSs, we first derived several possible rate expressions according to kinetics. To verify the above preliminary assumptions, the RDS was determined by adjusting the electrolyte pH, proton source, and CO partial pressure. Here, polycrystalline Cu made by sputter deposition was used as the model catalyst to investigate the RDS of COER to C$_{2+}$ products. After analysis those data, the hypothesized reaction mechanism was verified, and the data showed that the dimerization of two adsorbed *CO was the RDS. To exclude the potential effect of morphological or electronic state of Cu catalysts, similar experiments were conducted and the mechanism was approved to be the same over those catalysts.

## Results

### The theoretical kinetic derivations to guide mechanism verification

Based on previous DFT calculated and experimental mechanisms[13,21,22], this work first assumes possible RDSs of COER to C$_{2+}$ products. The (*)CO coupling (Table 1) and *CO protonation (Table 2) mechanisms were studied. Table 1 is further broken into two divisions: *CO-*CO coupling (step A2) or a *CO-CO(aq) coupling (step B2). In Table 2, the mechanism with the *CO protonation as the RDS is classified according to the source of the protons: H$_2$O (step C2), H$^+$ (step D2), *H from H$_2$O (step E3), or *H from H$^+$ (steps F3). By analyzing the specific RDS, we were able to speculate on the preceding and following reaction processes, which are detailed in Tables 1, 2 (See Supplementary Note 1 for more speculating detail).

Combining the different proposed mechanisms, the theoretical current density can be calculated by using the kinetic derivation as our previous work demonstrated[35]. The detailed derivation process can be found in the Supplementary Note 1. After obtaining the expression, theoretical parameters such as the reaction order of a specific reactant can be obtained from this expression, which can give guidance towards expected kinetic trends from experiments. Taking the RDS of

two *CO coupling (step A2) as an example, the theoretical rate expression is derived as follows

$$j_{C2+} = k_{A2}^0 \left( \frac{K_{{}^*CO}^0 H_{CO} P_{CO}[L]}{1 + K_{{}^*CO}^0 H_{CO} P_{CO}} \right)^2 \exp(-\alpha f \eta) \quad (1)$$

Where $j_{C2+}$ is the current density of C$_{2+}$ products; $k^0$ is the standard forward rate constant and A2 in $k_{A2}^0$ represents the $k^0$ of step A2; $K_{{}^*CO}^0$ is the standard equilibrium constant of CO adsorption; [L] is the concentration of surface reaction sites; $H_{CO}$ is the Henry's constant of CO gas; $P_{CO}$ is the pressure of CO gas; $\alpha$ is the transfer coefficient; $f = F/RT$, where $R$ is the ideal gas constant, $T$ is absolute temperature, and $F$ is the Faraday constant; $\eta$ is the overpotential for the cathodic reaction. In Eq. (1), the only adsorbate of any substantial coverage is assumed to be *CO. This assumption is consistent with the low coverage of *H on the Cu surface calculated by density functional theory (see Supplementary Note 2 for simulated *H coverages). According to Eq. (1), the reaction orders of H$^+$ ($n_{H^+}$) and H$_2$O ($n_{H_2O}$) are both 0. The $n_{CO}$ can be calculated by

$$n_{CO} \equiv P_{CO} \frac{\partial \ln j_{C2+}}{\partial P_{CO}} \quad (2)$$

Combining Eqs. (1) and (2) yields:

$$n_{CO} = \frac{2}{K_{{}^*CO}^0 H_{CO} P_{CO} + 1} \quad (3)$$

Thus, the $n_{CO}$ varies from 2 to 0 with increasing $P_{CO}$.

Similarly, the theoretical kinetic parameters corresponding to each mechanism can be deduced (Tables 1 and 2). Because different reaction mechanisms will correspond to different reaction orders, they can be later measured by experiments to verify the hypothetical mechanism and thus obtain the true RDS of COER to C$_{2+}$ products. Although Tafel analysis is commonly used to study the RDS, it is not considered in this work. It is difficult to obtain an accurate value because it is very sensitive to mass transfer and the transfer coefficient is unknown[36,37].

### The fabrication of Cu catalysts for COER

To elucidate the RDS of COER to $C_{2+}$ products, polycrystalline Cu was chosen as the model catalyst. Cu was deposited on Si(100) wafers by magnetron sputtering. According to the survey X-ray photoelectron spectroscopy (XPS) spectra, no signal of the substrate materials was found on the samples indicating that the substrate was totally covered by Cu (Supplementary Fig. 3a, b). X-ray diffraction (XRD) patterns (Supplementary Fig. 3c) show that these films have polycrystalline structures. Scanning electron microscopy (SEM) images (Supplementary Fig. 3d, e) show the Cu catalyst films are evenly distributed over the substrates. The model catalysts were then used to study the RDS. Since Cu catalysts in industrial applications are generally polycrystalline rather than single crystals, this also allows the experimental results of this model catalyst to be more generalizable.

The electrochemical activity measurements were conducted in a custom three-electrode cell, where the cathodic electrolyte is continuously bubbled with CO and then injected into the cathode by a peristaltic pump. The CO transfer limitation was greatly suppressed by the high flow rate of the electrolyte with a boundary layer thickness of 12 μm in our setup (Supplementary Fig. 4), which was beneficial for the following electrochemical kinetics study. For COER performance, $C_2H_4$, $CH_4$, $C_2H_6$, and $H_2$ were the main gas products; glycolaldehyde, acetate, ethylene glycol, ethanol, propionaldehyde, and n-propanol were the main liquid products. The total Faradaic efficiency is near 100% (with variation within 10%) in seven different electrolytes (Supplementary Fig. 5). The $CH_4$, $C_2H_4$, and $H_2$ current densities are consistent throughout one-hour tests (Supplementary Fig. 6), indicating the Cu catalyst is sufficiently stable for analyzing the RDS.

### pH dependency experiments to determine $n_{H^+}$

To ascertain the $n_{H^+}$, four electrolytes with varied pH were used. These electrolytes are 0.1 M KOH (pH 13), 0.1 M KHCO₃ (pH 9), 0.1 M KH₂PO₄ (pH 3, prepared by adding H₃PO₄ to KOH solution), and 0.05 M K₂SO₄ (pH 2, prepared by adding H₂SO₄ to KOH solution). By applying the same cation concentration for all electrolytes, concerns about the effects of the cation are eliminated[38–40]. However, it is challenging to rule out the impact of anions in this situation. Adjusting the pH of electrolytes with the same level of K⁺ ions cannot prevent the change of electrolyte anions. Out of these four anions, $H_2PO_4^-$ is strongly adsorbed over Cu and may also act as a proton source[41,42], which would bias the experimental results under acidic conditions. To avoid this, an acidic solution of weakly adsorbed and proton-free $SO_4^{2-}$ anion[43] was also chosen as a test electrolyte.

As shown in Fig. 1a, the current density of the $C_{2+}$ products remain relatively constant with decreasing pH at the same potential. This phenomenon is also observed for the specific $C_{2+}$ products: $C_2H_4$ (Fig. 1b), ethanol (Supplementary Fig. 7a), n-propanol (Supplementary Fig. 7b), and acetate (Supplementary Fig. 7c). These results indicates that the increase in proton concentration does not promote the formation of $C_{2+}$ products, thus the $n_{H^+}$ should be 0 or even negative.

On the contrary, the activity of the COER to $CH_4$ and hydrogen evolution reaction (HER) is enhanced as the pH decreases (Fig. 1c, d), which is consistent with the results from the literature[30,44,45]. More analysis can be found below Supplementary Fig. 7. Considering that protons are continuously consumed under COER (as well as the competing hydrogen evolution reaction, HER), the local pH will increase. By constructing a mass transfer model, it was discovered that while there is a slight variation in local pH in comparison to bulk pH, the trend of pH remains the same (Supplementary Fig. 8 and Supplementary Note 3). Therefore, the local pH shift does not affect the experiment's conclusion.

### KIE experiments to determine whether $H_2O$ is engaged in the RDS or its previous steps

It is important to acknowledge that in an aqueous system, $H_2O$ is commonly present at a substantial concentration as an electrolyte, and

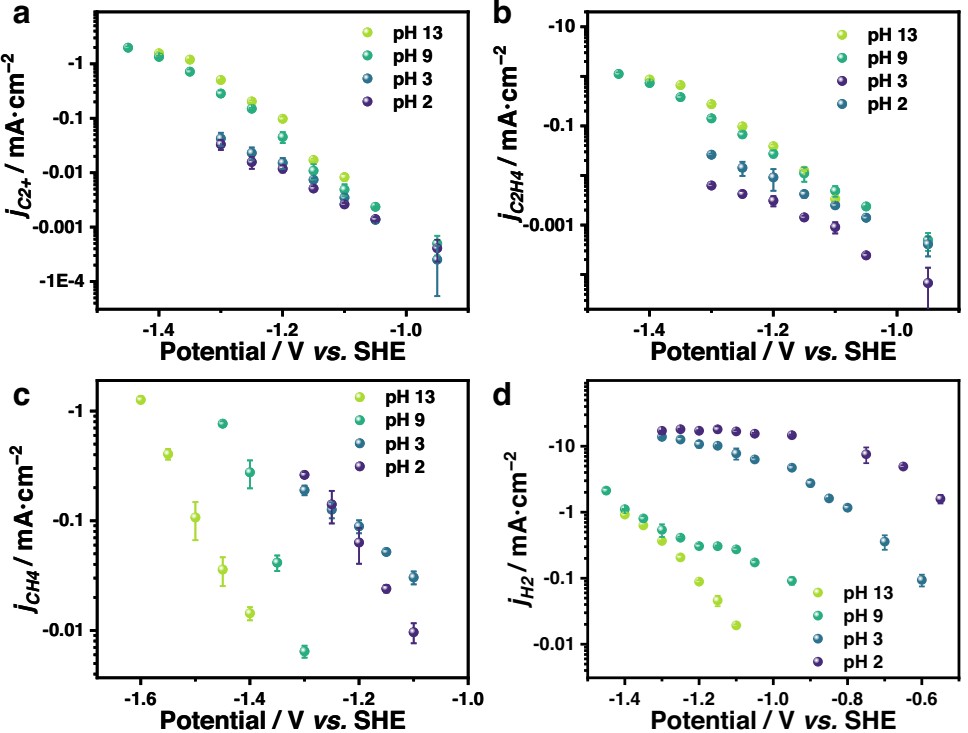

**Fig. 1 | The pH dependency for COER.** The current density of $C_{2+}$ (**a**), $C_2H_4$ (**b**), $CH_4$ (**c**), and $H_2$ (**d**) versus potential in CO-saturated electrolytes with different pH. All are measured in a potential range where the current is not too large in order to avoid getting into the limiting mass transport regime of CO. Error bars are means ± standard deviation (n = 3 replicates).

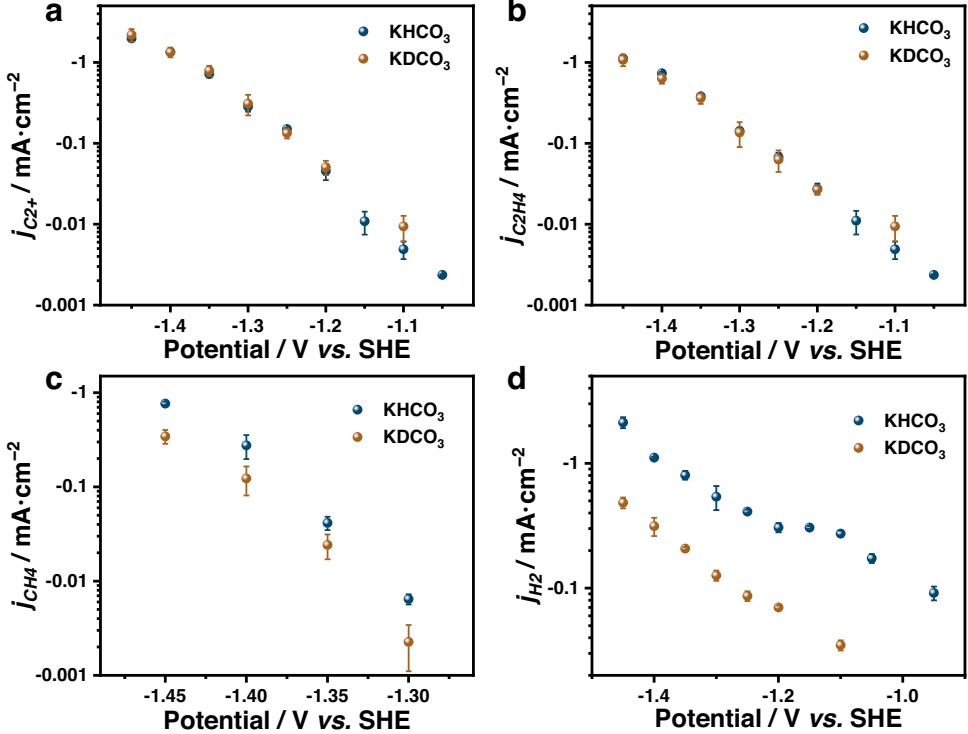

**Fig. 2 | Kinetic isotope effect for COER.** The current density of $C_{2+}$ (**a**), $C_2H_4$ (**b**), $CH_4$ (**c**), and $H_2$ (**d**) versus potential in CO-saturated 0.1 M KHCO$_3$ and KDCO$_3$ electrolytes. Error bars are means ± standard deviation ($n$ = 3 replicates).

its concentration remains constant throughout the reaction. Consequently, altering the $H_2O$ concentration does not enable us to determine its direct involvement in the reaction. However, the use of hydrogen/deuterium kinetic isotope exchange (H/D KIE) experiments can provide a means to differentiate whether $H_2O$ is engaged in the RDS of the reaction. The KIE experiments of COER in $D_2O$ (KDCO$_3$) and $H_2O$ (KHCO$_3$) solutions were conducted. From Fig. 2a, b, the current densities of $C_{2+}$ and $C_2H_4$ are identical under the two tested conditions, revealing that $H_2O$ is not involved either in the RDS or before the RDS. Supplementary Fig. 9 further illustrates the KIE effect for specific $C_{2+}$ products, including ethanol, n-propanol, and acetate. These results show that deuterium atoms can slightly increase ethanol generation while decreasing acetate production.

The current densities of $CH_4$ and $H_2$ were used as a benchmark of the KIE experiments. With the addition of deuterium atoms, the current densities of both $CH_4$ and $H_2$ dropped (Fig. 2c, d) in agreement with the fact that protons are involved in the RDS as reported in the literature[30,44,45]. The detailed explanations of the KIE can be found below Supplementary Fig. 10. Similarly, the KIE was tested in $H_2O$ and $D_2O$ solutions of KOH (pH 13) and $KH_2PO_4$ (pH 3) to exclude the potential effect of pH (Supplementary Fig. 10 and 11), where the same conclusion can be drawn. The previous literature reported that KIEs of H/D over various different copper surface can vary from 2 to 1[46]. It is conceivable that the surface state of the catalyst may be a determining factor in altering the RDS. The mechanistic system research method proposed in this work is expected to effectively guide future explorations of RDS in other Cu-based catalyst reactions.

**CO partial pressure effects to determine $n_{CO}$**
The partial pressure of CO (0.02 to 1 bar) was adjusted by changing the ratio of CO to Ar gas. All the experiments were conducted in a KOH electrolyte at −1.3 V *vs*. SHE. Prior to conducting the reaction kinetics analysis, the conversion rate of CO under different partial pressures was calculated to be approximately 0.02% (Supplementary Fig. 12).

This confirms that there will be sufficient CO supply at low pressures. Figure 3a shows that the activity of $C_{2+}$ products rises along with the CO partial pressure until it reaches 0.5 bar, after which the partial current density becomes constant. By using a simple linear fitting method, the $n_{CO}$ is roughly equivalent to 1 at partial pressures lower than 0.15 bar and 0 at partial pressures higher than 0.5 bar, which are consistent with data reported in the literature[21,30,31]. However, a simple interval linear fit, without the possibility of extrapolation, is likely to lead to erroneous conclusions that $n_{CO}$ is between 1 ~ 0 from low to high CO pressure.

In order to be more accurate, the theoretical rate expression is employed as the fitting formula (see Supplementary Note 4 for more fitting detail). It was determined that only the mechanism of step A2 (*CO-*CO coupling as the RDS) can fit the experimental data well (Fig. 3 and Supplementary Fig. 13–16). To exclude the influence of pH, similar CO partial pressure experiments were measured in KOH electrolyte under alkaline condition at −1.3 V *vs*. SHE. The same conclusion can be obtained (Supplementary Fig. 17).

When discussing $n_{CO}$, there are one thing to keep in mind. The data must first be fitted according to the theoretical rate expression; otherwise, it may result in incorrect conclusions due to the lack of suitable low or high CO pressure data. For example, $n_{CO}$ is assumed to be between 1 and 0 by a simple interval linear fit, but after the non-linear fitting, only step A2 is found to be consistent with the experimental data.

## Discussion
The mechanisms in Tables 1, 2 were subsequently verified by the previous experimental results, which confirmed the reaction orders of CO and $H^+$ are 0 and ≤0, respectively at a CO pressure of 1 bar. Moreover, $H_2O$ does not participate in the RDS or any preceding steps at a CO pressure of 1 bar. Out of the above-mentioned hypothesized processes, only the mechanism of *CO-*CO coupling (step A2) as the RDS satisfies the experimental findings at the same time.

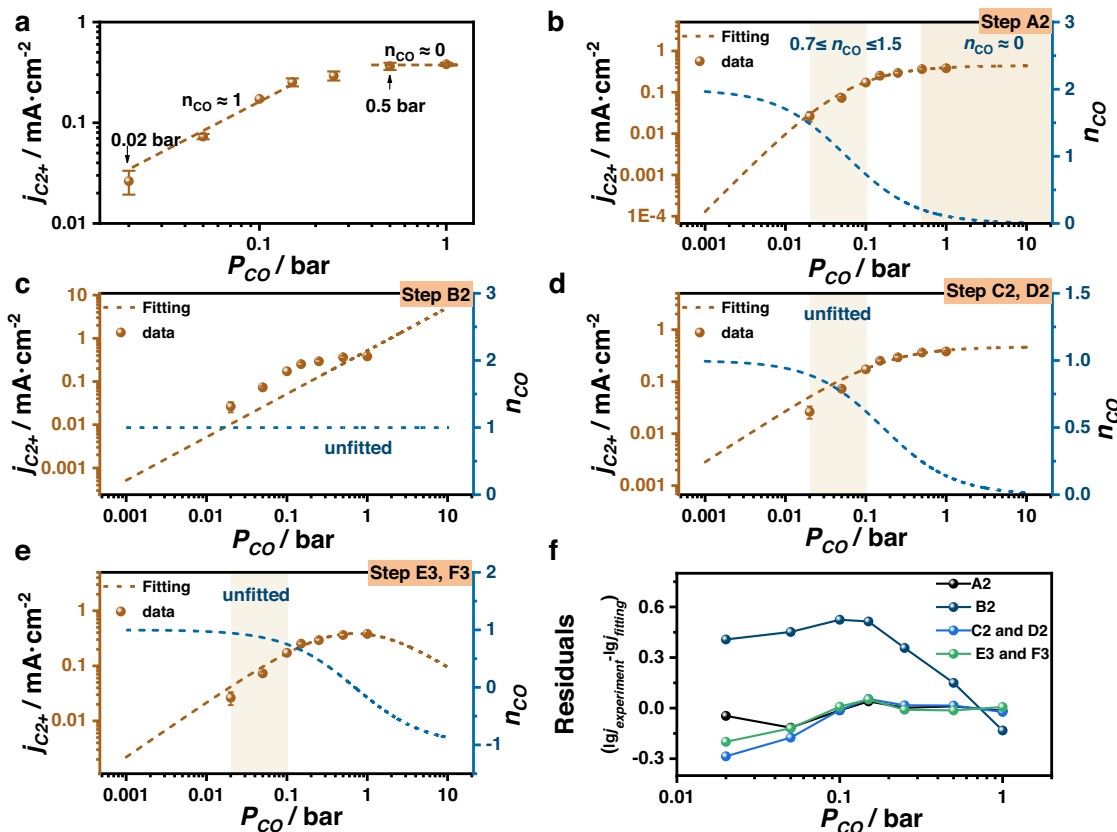

**Fig. 3 | CO reaction order for COER.** The current density of $C_{2+}$ products versus CO partial pressure for interval linear (**a**) and non-linear (**b**–**e**) data fitting as well as their residuals (**f**). The equations used for fitting (**b**–**e**) are the theoretical rate expressions corresponding to different mechanisms (see Supplementary Table 1 for all the fitting parameters). All those data are collected in CO-saturated 0.1 M KOH electrolyte at −1.3 V vs. SHE. Error bars are means ± standard deviation ($n = 3$ replicates).

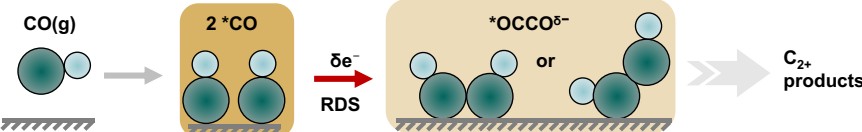

**Fig. 4 | Reaction mechanism of $C_{2+}$ products diagram for COER on polycrystalline Cu.** The CO first adsorbs over Cu catalysts, following two adsorbed CO coupling as the RDS of $C_{2+}$ products.

To further demonstrate the generalizability of the conclusions, the investigations in other Cu-based catalytic reactions, commercial Cu nanoparticles (Cu NP 40–60 nm) and oxide-derived Cu (OD-Cu), was carried out. It can also be observed that the activity of $C_{2+}$ products does not increase with higher $H^+$ concentration (Supplementary Fig. 18–21). The KIE was approximately equal to 1, and the partial pressure experiment of CO could only be fully fitted using the theoretical formula for Step 2 (*CO-*CO coupling). Thus, for both commercial Cu and OD-Cu, the RDS for $C_{2+}$ products should also be the *CO-*CO coupling.

It is worth mentioning that our primary experiments were conducted under both acidic and alkaline conditions. Therefore, our conclusions should not vary with pH. However, the activity of ethanol, acetate, and n-propanol products was observed to be very low. As a result, the experimental data primarily reflected the contribution of $C_2H_4$. Therefore, the conclusions drawn from this data are more applicable to $C_2H_4$, and it cannot be ruled out that specific $C_{2+}$ products may have different RDS than those mentioned in the article. In the future, the research of a particular product or $C_{3+}$ products should

be conducted. To investigate this, it would be necessary to identify a catalyst that favors the production of this product and then employ similar kinetic analysis methods as used in this work. By considering various possibilities, designing experiments, and finally validating hypotheses, it would be possible to unravel the RDS for specific $C_{2+}$ products formation.

In conclusion, this work combines theoretical derivation with hypothesized mechanisms to design a method that can effectively verify the mechanism of the RDS of COER to $C_{2+}$ products. It provides comprehensive kinetic data through experiments, including pH dependency, KIE, and CO partial pressure experiments. Specifically, the reaction is not accelerated by an increase in proton concentration, it is not slowed down by the replacement of deuterium atoms for $H^1$, and the $C_{2+}$ reactivity rises initially before remaining constant as CO partial pressure increases. Verifying these hypothetical reaction mechanisms according to the experiment data, it was found that only the dimerization of two adsorbed *CO is the most likely RDS (Fig. 4). This finding suggests that promoting the C-C coupling is key to enhancing $C_{2+}$ products.

## Methods

### Electrode preparation

Cu thin films were deposited onto single-crystal Si wafers with the (100) orientation using an AJA ATC Orion-5 magnetron sputtering system. To enhance the adhesion between the catalysts and the Si wafers, Ti films with a thickness of approximately 15 nm were first deposited on the Si wafers[47]. In detail, the Si wafers were etched with $Ar^+$ ions for 5 min with a power of 40 W to clean the silicon oxide on Si wafers. Then, 15 nm Ti films (99.9999%) were deposited as binders between catalysts and Si wafers at the power of 100 W. Cu catalyst films (99.9999%) with a thickness of 100 nm were deposited over Ti at 130 W. For commercial Cu nanoparticle was purchase from Sigma-Aldrich with 40 - 60 nm size. OD-Cu is prepared by subjecting commercial Cu nanoparticles to a reduction process under in-situ negative potential after being treated in an $O_2$ atmosphere at 300 °C for 1 hour.

### Electrode characterization

The crystal structures of the Cu thin films were analyzed with a Rigaku Smartlab X-ray diffractometer (XRD) using Cu Kα radiation (40 kV, 40 mA). The near-surface compositions of the thin films were measured with a Kratos Axis Ultra DLD X-ray photoelectron spectrometer (XPS). All spectra were acquired using monochromatized Al Kα radiation (15 kV, 15 mA). The kinetic energy scale of the measured spectra was calibrated by setting the C 1 s binding energy to 284.8 eV. The surface structure of those thin films was recorded using an FEI XL30 Sirion scanning electron microscope (SEM) at the acceleration voltage of 5 kV, ET-detector, SE mode.

### Chemicals

Potassium hydroxide hydrate (KOH·H₂O, Merck, 99.995%), potassium carbonate ($K_2CO_3$, Merck, 99.995%), phosphoric acid ($H_3PO_4$, Sigma Aldrich, 85 wt% in $H_2O$, 99.99%), sulfuric acid ($H_2SO_4$, Sigma Aldrich, 99.999%), deuterium oxide ($D_2O$, Sigma Aldrich, 99.9 atom % D).

### Preparing electrolyte

Seven different electrolytes were used in the work, including 0.1 M KOH (pH 13), 0.1 M KHCO₃ (pH 9, the electrolyte was made by bubbling $CO_2$ into 0.05 M $K_2CO_3$ electrolyte for 4 hours and then bubbling Ar overnight to get rid of excess $CO_2$), 0.1 M $KH_2PO_4$ (pH 3, the electrolyte was made by neutralizing KOH with $H_3PO_4$ to pH 3, and the $K^+$ concentration in the electrolyte was kept at 0.1 M), 0.05 M $K_2SO_4$ with $H_2SO_4$ (pH 2, the electrolyte was made by neutralizing KOH with $H_2SO_4$ to pH 2, and the $K^+$ concentration in the electrolyte was kept at 0.1 M), 0.1 M KOH in $D_2O$, 0.1 M $KDCO_3$ (the electrolyte was made by the same method as 0.1 M KHCO₃ except the $H_2O$ was changed to $D_2O$), and 0.1 M $KH_2PO_4$ in $D_2O$ (the electrolyte was made by the same method as 0.1 M $KH_2PO_4$ except the $H_2O$ was changed to $D_2O$). All the solutions were prepared using 18.2 milli-Q water (Synergy UV) or deuterium oxide ($D_2O$, Sigma Aldrich, 99.9 atom % D).

### Electrochemical characterization

Most electrochemical activity measurements were conducted in a custom electrochemical cell machined from PEEK at room temperature and atmospheric pressure (Supplementary Fig. 4a). The cell was sonicated in 20 wt% nitric acid and thoroughly rinsed with milli-Q water before all experimentation. The working and counter electrodes were parallel and separated by a bipolar membrane (BPM, Fumasep FBM). The exposed geometric surface area of each electrode was 1 cm². The electrolyte volumes in the cathodic and anodic chambers were 7 mL and 0.9 mL, respectively. The counter electrode was iridium dioxide ($IrO_2$) purchased from Dioxide Materials. The working electrode potential was referenced against a Hg/HgO electrode in 0.1 M KOH (ALVATEK, RE-61AP) that was calibrated against a homemade standard hydrogen electrode (SHE). The solutions mentioned before were used as the cathodic electrolyte. 0.1 M KHCO₃ was used as the

anodic electrolyte. The cathodic electrolyte was sparged with CO (99.999% Praxair Inc.) or mixture with Ar (99.999% Praxair Inc.) at a certain rate, where the 10 sccm of CO was used for pH-dependence and KIE experiments, and 20 sccm of different ratio of CO and Ar was used for partial pressure experiments. Note that CO gas must be purified by using a carbonyl trap (LPM Carbonyl Trap) in order to eliminate metal ions as Ni, Fe, etc. from metal tubes. Then gas-saturated electrolyte was pumped into the cathodic chamber by using a peristaltic pump (SHENCHEN LabN6) with the rate of 100 rev/min. Here, the experiment process in this pump speed cannot be significantly diffusion-limited, since we have obtained relatively straight Tafel slopes over 3 orders of magnitude in the current (Figs. 1, 2).

For the flow cell experiments, they were performed in three-component cell configuration. For specific flow cell details, please refer to our previous the article[48]. A constant CO flowrate of 30 mL·min⁻¹ was purged into gas compartment, and 30 mL catholyte and anolyte were applied, respectively. A part of CO diffused to the catalyst surface in electrolyte for CO conversion, forming various gaseous and liquid products. The gaseous products mixed with unreacted CO were vented out of the electrolyzer, injecting into the gas-sampling loop of a gas chromatography for identification and quantification. The liquid products were measured from both catholyte and anolyte.

The produced CO and $H_2$ are tested by gas chromatography (GC, Thermo scientific, TRACE 1300). Ar was used as the carrier gas. The GC was equipped with a packed Molsieve 5 A column, a packed Hayesep Q column, and an Rt-Qbond column to separate the gaseous products. Thus, $H_2$ and carbon-containing products ($CH_4$, $C_2H_4$, $C_2H_6$) could be identified using a thermal conductivity detector and a flame ionization detector, respectively. The liquid-phase products are analyzed after the electrolysis using high-performance liquid chromatography (HPLC, Agilent 1200 series). Liquid-phase products were separated by an Aminex HPX-87H column (Bio-Rad) that was maintained at 50 °C. The HPLC was equipped with a a refractive index detector (RID). 5 mM $H_2SO_4$ solution was used as the carrier liquid with 0.3 mL min⁻¹ speed. Some raw HPLC sample data were shown in Supplementary Fig. 22. The response signals of the RID were calibrated by solutions with different concentrations (Supplementary Fig. 23).

Electrochemical characterizations were performed using a Biologic VSP-300 potentiostat. All electrochemical measurements were recorded versus the reference electrode and converted to the SHE scales. IR compensation (ZIR) was used to determine the uncompensated resistance ($R_u$) of the electrochemical cell.

The Cu catalyst was firstly reduced by conducting chronoamperometry at −0.9 V vs. SHE for 10 min. Under this potential, no carbon-related products can be produced. The electrocatalytic activity of the Cu catalyst was assessed by conducting chronoamperometry at different potentials for 1 hour. The gas product was measured every 10 min, and the liquid product was measured after one-hour experiments. Each data was tested at least three separate times to ensure the statistical relevance of the observed trends.

## Data availability

All the data that support the findings of this study are available within the paper and its Supplementary Information files, or from the corresponding author on reasonable request.

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

## Acknowledgements

This work was supported by the National Key R&D Program of China (2021YFA1501503), the National Natural Science Foundation of China (22121004, 22038009, 22108197, 22250008), the Haihe Laboratory of Sustainable Chemical Transformations (CYZC202107), the Program of Introducing Talents of Discipline to Universities (No. BP0618007), the Xplorer Prize, the European Union's Horizon 2020 research, innovation program under grant agreement No. 851441 (project SelectCO2), Villum Center for the Science of Sustainable Fuels and Chemicals (V-Sustain No. 9455).

## Author contributions

W.Y.D. designed and performed the experiments. Y.Q. is responsible for collecting XPS and XRD data. G.K. and N.G. are responsible for the calculation of the H coverage. A.N.X helped the calculation of local pH. P.Z., B.S., I.C. and J.L.G. coordinated and supervised the research. All authors contributed to the writing of the manuscript.

## Competing interests

The authors declare no competing interests.
