## [Peer Review File · Nature Communications]

Reviewers' comments:

Reviewer #3 (Remarks to the Author):

The authors made revisions and corrected some errors according to my comments. Some major issues are still pending to be addressed. The analysis methods and main conclusion are similar to those reported in the literature (ACS Catal. 2022, 12, 9, 5275–5283; Nat. Commun. 2021, 12, 3264; Energy&Fuels 2023, 37, 11, 7904–7910). Thus, no new insight was provided by this work. In addition, it is not logical how the trend of a specific C₂⁺ product may vary with pH or H/D substitution when the total C₂⁺ activity remains relatively constant. The authors claim that their catalysts are selective for ethylene. Actually, many oxide-derived catalysts were reported for selective oxygenate formation. These catalysts can be used to probe the reaction order of C₂⁺ liquid products.

Regarding the response to comment 2, the figures given support my comments in the previous question. The activity toward ethylene and ethanol decreases with pH, but the activity for acetate and propanol increases. I agree with the response that after undergoing the same RDS, a higher pH favors ketene obtaining OH⁻, resulting in the observation of more Acetate, while the activity of ethylene and ethanol decreases. But what about propanol?

Minor comments:

1. Response to comment 5 is acceptable. But has ethanol been reported to form via the reduction of ketene intermediate?
2. Why was -1.5V vs. SHE used in the flow cell? Following the authors' explanation, a low potential in the flow cell should also work.

Reviewer #4 (Remarks to the Author):

In the context of developing sustainable fuel and energy, CO electrochemical reduction is an important process to achieve this goal. In this work, the authors explored the rate-determining steps of the generation of multi-carbon products in this process through a series of experiments, solved the problem of unclear reaction mechanism of this process, and laid an important experimental and theoretical foundation for improving the yield of C₂⁺ products in the future.

The authors have given detailed and reasonable answers to the reviewers' comments (especially the reviewer 1#) and improved the quality of the manuscript. Therefore, I think the revised manuscript can be accepted by Nature Communication at this stage.

Reviewer 3:

General Comments R3: *The authors made revisions and corrected some errors according to my comments. Some major issues are still pending to be addressed. The analysis methods and main conclusion are similar to those reported in the literature (ACS Catal. 2022, 12, 9, 5275–5283; Nat. Commun. 2021, 12, 3264; Energy&Fuels 2023, 37, 11, 7904–7910). Thus, no new insight was provided by this work.*

In addition, it is not logical how the trend of a specific C₂₊ product may vary with pH or H/D substitution when the total C₂₊ activity remains relatively constant.

The authors claim that their catalysts are selective for ethylene. Actually, many oxide-derive catalysts were reported for selective oxygenate formation. These catalysts can be used to probe the reaction order of C₂₊ liquid products.

Regarding the response to comment 2, the figures given support my comments in the previous question. The activity toward ethylene and ethanol decreases with pH, but the activity for acetate and propanol increases. I agree with the response that after undergoing the same RDS, a higher pH favors ketene obtaining OH⁻, resulting in the observation of more Acetate, while the activity of ethylene and ethanol decreases. But what about propanol?

Response: We sincerely appreciate the referee for the valuable feedback and comments on our work. We have carefully considered all the suggestions and have addressed the issues.

i) no new insight was provided by this work.

We appreciate your thoughtful review of our manuscript, and we would like to address your thinking regarding the similarity of our analysis methods and main conclusions to those reported in the literature (ACS Catal. 2022, 12, 9, 5275–5283; Nat. Commun. 2021, 12, 3264; Energy&Fuels 2023, 37, 11, 7904–7910). While we acknowledge that the field of CO electroreduction and the study of RDS for C-C coupling have been actively researched, it is important to note that our work contributes unique insights and advancements to the field.

As you can see from Table 1, the similarity with ACS Catal. 2022, 12, 9, 5275–5283 lies in our common approach of CO reaction order study, while the RDS of C₂₊ cannot be conclusively determined solely from this experiment, and furthermore, this article does not investigate the RDS of C₂₊.

For the *Nat. Commun. 2021, 12, 3264*, they only conducted pH experiments under alkaline conditions. However, H₂O acts as the proton source rather than H⁺ under such conditions. In subsequent work by their research group (*Angew. Chem. Int. Ed.*, 2022, 61, e202111167), they also noted this aspect. However, due to incomplete kinetic analysis, they drew entirely different conclusions (Table 2).

For the *Energy&Fuels 2023, 37, 11, 7904–7910*, they provided an understanding of reaction kinetics leading to improved performance toward specific products by kinetic experiments. While the article didn't focus on the RDS of C₂₊, no rigorous kinetic analysis of RDS is provided in this paper.

Our work employs a comprehensive kinetic experimental design, as evidenced by the comparison with similar articles in this field. Notably, our utilization of nonlinear fitting methods for analyzing the reaction order of CO has led to entirely different conclusions, even when using similar experimental data. This highlights the significance of data analysis methodologies in this particular field.

Table 1. The comparison of methods and conclusions in previous works and our work.

	Methods	Differences	Conclusions
ACS Catal. 2022, 12, 9, 5275–5283	Kinetic experiments: CO reaction order Tafel slope	No RDS mechanism is explored	The reaction pathway is *CO-*CHO
Nat. Commun. 2021, 12, 3264	Kinetic experiments: pH (7-14)	It only conducted pH experiments under alkaline conditions. However, water (H₂O) acts as the proton source rather than H⁺ under such conditions.	*CO-*CO coupling is the RDS
Energy&Fuels 2023, 37, 11, 7904–7910	Kinetic experiments: CO reaction order KIE pH (7.2-14.2)	1. Only conducted pH experiments under alkaline conditions. 2. Lack of rigorous kinetic analysis of RDS (CO Reaction order). Didn't focus on RDS of C₂+	Provide the understanding of reaction kinetics leading to improved performance toward specific products.
Our work	Kinetic experiments: CO reaction order KIE pH (2-13)	Nonlinear fitting of CO reaction order and pH dependency experiment in both acidic and alkaline conditions	*CO-*CO coupling is the RDS

Table 2. The comparison of methods and conclusions in previous works and our work.

	Methods	Differences	Conclusions
Angew. Chem. Int. Ed., 2022, 61, e202111167	Kinetic experiments: CO reaction order KIE	Incorrect analysis of the reaction order of CO has led to erroneous conclusions.	*CO+H ₂ O is the RDS (Different conclusions were reached by the same group below)
Nat. Commun. 2021, 12, 3264	Kinetic experiments: pH (7-14)	It only conducted pH experiments under alkaline conditions. However, water (H ₂ O) acts as the proton source rather than H ⁺ under such conditions.	*CO-*CO coupling is the RDS
Our work	Kinetic experiments: CO reaction order KIE pH (2-13)	Nonlinear fitting of CO reaction order and pH dependency experiment in both acidic and alkaline conditions	*CO-*CO coupling is the RDS

ii) In addition, it is not logical how the trend of a specific C₂₊ product may vary with pH or H/D substitution when the total C₂₊ activity remains relatively constant.

Regarding the response to comment 2, the figures given support my comments in the previous question. The activity toward ethylene and ethanol decreases with pH, but the activity for acetate and propanol increases. I agree with the response that after undergoing the same RDS, a higher pH favors ketene obtaining OH⁻, resulting in the observation of more Acetate, while the activity of ethylene and ethanol decreases. But what about propanol?

Thank you for your insightful comments regarding the trend of a specific C₂₊ product's variation with pH or H/D substitution while the total C₂₊ activity remains relatively constant. When considering specific C₂₊ products such as C₂H₄, ethanol, acetate, or n-propanol, the influence of H⁺ on the selectivity after the RDS cannot be avoided. Furthermore, as the liquid products are not immediately removed and stay in the electrolyte until the end of the experiment, they could further react either electrocatalytically or homogeneously. Consequently, it is difficult to determine the RDS for a specific product by conducting kinetic analysis on the catalyst when it does not prefer to produce such a product.

Regarding your question about propanol, the activity of propanol seems relatively consistent in different pH (Supplementary Fig. 18b and d). The trends shown in Fig. R1b also fall within the range of experimental error, making it challenging to draw the conclusion that the propanol activity increases with pH. We concur with the reviewer's suggestion that future work may involve the investigation of catalysts with high selectivity for specific C₂₊ products, which could provide better insights into the RDS for those particular products.

Fig. R1. The current density of different products at different pH for commercial Cu nanoparticles (Cu NP 40~60 nm, Sigma-Aldrich) at -1.5 V vs. SHE in flow cell. All those data are collected in CO-saturated 1 M KOH electrolyte (pH 14) and 1 M KHCO₃ (pH 8.5).

Supplementary Fig. 18. The current density of different products at different pH for (a and b) commercial Cu nanoparticles (Cu NP 40~60 nm, Sigma-Aldrich) and (c and d) oxide-derived Cu (OD-Cu) at -1.5 V vs. SHE in a flow cell. The application of a flow cell in those experiments is primarily motivated by two considerations. Firstly, it is difficult to measure this type of powdered catalyst in our H-cell, as it is challenging to prepare such catalyst films. Additionally, it is interesting to investigate whether the mechanism would be different under higher current density or different cell configurations. For specific flow cell details, please refer to our previous article.¹⁸ OD-Cu is prepared by subjecting commercial Cu nanoparticles to a reduction process under in-situ negative potential after being treated in an O_2 atmosphere at 300 °C for 1 hour.

iii) The authors claim that their catalysts are selective for ethylene. Actually, many oxide-derive catalysts were reported for selective oxygenate formation. These catalysts can be used to probe the reaction order of C_2^+ liquid products.

We appreciate your valuable comment on our manuscript. We acknowledge that oxide-derived catalysts have indeed been reported for selective oxygenate formation, and we thank you for pointing this out. In the supplementary experiments, the results with OD-Cu catalyst indeed indicate a preference for oxygenate formation (Supplementary Fig. 18). However, achieving exceptionally high selectivity remains challenging. While investigating the oxygenate reaction mechanism in future work, it is indeed essential to choose catalysts with high selectivity. However, given that this is not the primary focus of our current study, we will refrain from delving further into this discussion.

Specific Comment R3-1: *Response to comment 5 is acceptable. But has ethanol been reported to form via the reduction of ketene intermediate?*

Response: Thank you for your positive feedback regarding comment 5, which we appreciate. Regarding your query about the formation of ethanol via the reduction of a ketene intermediate, we would like to provide clarification. According to the literature, ketene, when adsorbed on a surface, can undergo further hydrogenation to produce ethylene and ethanol, and upon desorption, it may be attacked by OH⁻ to generate acetate (*Nat. Commun.* **2019**, *10* (1), 32; *Energy Environ. Sci.* **2022**, *15* (9), 3978-3990). Therefore, some literature does think ketene could be the intermediate of ethanol.

Specific Comment R3-2: *Why was -1.5V vs. SHE used in the flow cell? Following the authors' explanation, a low potential in the flow cell should also work.*

Response: Thank you for your inquiry regarding the choice of -1.5V vs. SHE in our flow cell setup. Considering that a higher potential is required for CH₄ generation and that the same voltage was employed in our previous CO reaction order experiments in the H-cell, we have chosen this potential here. We have added the further explanation in the manuscript. (*Page S35*)

Page S35:

Considering that a higher potential is required for CH₄ generation and that the same voltage was employed in our previous CO reaction order experiments in H-cell, -1.5 V vs. SHE was applied here.

Reviewer 4:

General Comments R3: *In the context of developing sustainable fuel and energy, CO electrochemical reduction is an important process to achieve this goal. In this work, the authors explored the rate-determining steps of the generation of multi-carbon products in this process through a series of experiments, solved the problem of unclear reaction mechanism of this process, and laid an important experimental and theoretical foundation for improving the yield of C₂+ products in the future.*

The authors have given detailed and reasonable answers to the reviewers' comments (especially the reviewer 1#) and improved the quality of the manuscript. Therefore, I think the revised manuscript can be accepted by Nature Communication at this stage.

Response: We sincerely appreciate the referee for the positive comment on the novelty and depth of insights of our work.